# Convolutional Neural Network for Breathing Phase Detection in Lung Sounds

**DOI:** 10.3390/s19081798

**Published:** 2019-04-15

**Authors:** Cristina Jácome, Johan Ravn, Einar Holsbø, Juan Carlos Aviles-Solis, Hasse Melbye, Lars Ailo Bongo

**Affiliations:** 1CINTESIS-Center for Health Technologies and Information Systems Research, Faculty of Medicine, University of Porto, 4200-450 Porto, Portugal; cjacome@med.up.pt; 2Medsensio AS, N-9037 Tromsø, Norway; johan@medsens.io; 3Department of Computer Science, UiT The Arctic University of Norway, N-9037 Tromsø, Norway; einar.j.holsbo@uit.no; 4General Practice Research Unit in Tromsø, Department of Community Medicine, UiT The Arctic University of Norway, N-9037 Tromsø, Norway; juan.c.solis@uit.no (J.C.A.-S.); hasse.melbye@uit.no (H.M.)

**Keywords:** respiratory phases, breath onset, breath detection, spectrograms, automated classification, deep learning

## Abstract

We applied deep learning to create an algorithm for breathing phase detection in lung sound recordings, and we compared the breathing phases detected by the algorithm and manually annotated by two experienced lung sound researchers. Our algorithm uses a convolutional neural network with spectrograms as the features, removing the need to specify features explicitly. We trained and evaluated the algorithm using three subsets that are larger than previously seen in the literature. We evaluated the performance of the method using two methods. First, discrete count of agreed breathing phases (using 50% overlap between a pair of boxes), shows a mean agreement with lung sound experts of 97% for inspiration and 87% for expiration. Second, the fraction of time of agreement (in seconds) gives higher pseudo-kappa values for inspiration (0.73–0.88) than expiration (0.63–0.84), showing an average sensitivity of 97% and an average specificity of 84%. With both evaluation methods, the agreement between the annotators and the algorithm shows human level performance for the algorithm. The developed algorithm is valid for detecting breathing phases in lung sound recordings.

## 1. Introduction

Lung auscultation is an important part of routine physical examinations [1]. During auscultation, clinicians assess the presence of normal and adventitious lung sounds (e.g., crackles, wheezes) generated by the air flow in the respiratory tract which can be correlated with lung mechanics including movement of air, changes within lung morphology, and presence of secretions. Besides monitoring the presence of normal and adventitious sounds, clinicians need to be aware of their timing in the respiratory cycle (early/mid/late inspiratory or expiratory) as it may have clinical significance for the assessment of patient respiratory status and for the differential diagnosis of cardiorespiratory disorders [2]. For example, fine crackles in mid-to-late inspiration are associated with interstitial lung fibrosis, congestive heart failure, pneumonia; while coarse crackles, that appear early during inspiration and throughout expiration, are associated with chronic bronchitis [1].

The development of digital stethoscopes and computerized techniques have enabled the recording and the automatic, real-time analysis of lung sounds with minimal setup, enabling the characterization of the power spectra of normal lung sounds and the identification and quantification of adventitious lung sounds. However, until recently, less attention has been given to the development of techniques for the automatic detection of breathing phases (inspiration and expiration) from lung sound recordings, which is needed for real-time identification of the timing of each auscultation finding. To deal with this limitation, researchers have combined airflow measured simultaneously with lung sound recordings [3]. Nevertheless, this strategy demands a complex setup and it is not compatible to clinical practice needs, nor applicable for smart stethoscopes.

To address this restraint, some signal processing methods have been proposed in the past years to detect breathing phases directly from lung sound recordings. Chuah and Moussavi (2000) proposed a method using the average chest power spectra to detect the breathing phases and the average tracheal power spectra to determine the breath onsets. They validated their method in 11 healthy subjects and found a phase detection accuracy of 93% and a breath onset detection accuracy of 100% [4]. Later, Huq and Moussavi (2012) developed an automatic method for breath phase detection using only tracheal sounds and validated it in 93 healthy subjects [5]. This method was based on several breath sound parameters (peak intensity, duration, etc.) and showed an accuracy of 95.6% for breath-phase identification. More recently, Reyes et al. (2016) proposed the automatic classification of breathing phases from a smartphone optical recording of the chest movements acquired simultaneously with tracheal sounds. This was tested in 13 healthy adults and a 100% accuracy was found [6]. However, these methods were heavily dependent on tracheal sounds and it is well known that the characteristics of tracheal and chest sounds are distinct. Tracheal sounds contain high frequencies and are easily heard during the two breathing phases, which is mainly related to trachea large diameter and the absence of a structure to filter the sound [1,7]. Conversely, chest sounds are normally heard clearly during inspiration but only in the early phase of expiration, as a result of the flow becoming laminar and the high frequencies being filtered by the lung parenchyma [1,7]. Moreover, to monitor cardiorespiratory diseases, tracheal auscultation is not frequently performed as it mainly reflects the status of the upper airways [1].

In addition, the above methods were developed and tested on small datasets of healthy subjects. Breathing pattern and lung sounds are known to change in the presence of respiratory diseases [8,9] and thus these methods may not be applicable to subjects with respiratory diseases.

Novel approaches to detect breathing phases using lung sounds from typical auscultation sites (e.g., posterior chest), from a larger dataset of subjects with or without cardiorespiratory diseases should therefore be explored. In recent years, the introduction of neural networks has improved acoustic signals source identification, such as speech recognition [10] and has been shown to rival human level classification performance [11]. Speech recognition traditionally uses Mel Frequency Cepstral Coefficients (MFCC’s), but recently spectrograms, containing intensity information of time varying spectrum of a waveform, have been shown to outperform MFCC’s [12]. To our best knowledge, neural networks using spectrograms as features have not been previously used to detect breathing phases.

Thus, the aim of this study was to develop and evaluate the performance of an algorithm to detect breathing phases based on spectrograms using lung sounds from subjects with or without cardiorespiratory diseases.

## 2. Materials and Methods

### 2.1. Data Sets

We used a sample of lung sound files from adults participating in the Tromsø 7 study [13]. The Tromsø study, initiated in 1974, is a longitudinal, multipurpose, population-based Norwegian study of health conditions and chronic diseases conducted every 6–7 years in the Tromsø municipality. For the Tromsø 7 study in 2015–2016, 21,083 participants attended for a first visit (65% of the invited) and 6,048 (mean age 63.2 years, 54.7% female) had their lung sounds recorded. Because of the high attendance rate of the study, we believe that our random sample is representative for its age group in the area.

Lung sounds were recorded using an electret microphone (MKE 2-eW Gold, Sennheiser electronic GmbH & Co. KG) inserted at the tube of a stethoscope, 10 cm away from the chest piece. The microphone was tuned to a sensitivity of −12 dB. The sound files were captured in Wave (.wav) format at 44,100 Hz sampling rate. The audio files were not further processed after recording. The patients breathed in and out with an open mouth and deeper than normal. Sounds were recorded at six chest locations, three on each side of the chest (on the back between the spine and the medial border of the scapula at the level of T4–T5; at the middle point between the spine and the mid axillary line at the level of T9–T10; and on the front where the medioclavicular line crosses the second rib) during 10 or 15 s (10 s if a second recording was made in the same chest location).

In this study, three subsets from the Tromsø 7 lung sound dataset were used: subsets one and two for training, and subset three for evaluation (Table 1). The target was to train the algorithm to detect breathing phases acquired at 6 distinct chest locations, so we aimed to have approximately 200 recordings per chest location and of distinct durations (Subset 1 and 2). For evaluation, we aimed to have at least 10% of the size of the training subset (Subset 3). Subset one consisted of 1022 files with 10 s from 85 subjects (mean age 59.9 year; 55.3% female). Subset two contained 112 files with 15 s (mean age 64.3 years, 44% female). The third subset consisted of 120 sound files with 15 s from 20 randomly selected subjects (mean age 68.3 years; 65% female).

### 2.2. Manual Annotation of Breathing Phases

The breathing phases from training subset 1 were manually annotated by a physiotherapist/lung sound researcher (C. Jácome, Annotator 1) using the Respiratory Sound Annotation Software [14]. Using this tool, the annotator listens to the sound while visualizing its waveform and identifies the onset and end of each breathing phase. A total of 3212 inspiration phases and 2842 expiration phases were identified. The breathing phases of the second subset were identified by a first version of our algorithm (trained on subset 1) and were visually inspected and corrected by a computer scientist with previous experience on detection of wheezes in lung sounds (J. Ravn, Annotator 2). This subset, containing only 112 files, was much smaller than subset 1 but the files were longer, 15 s. The use of 2 training subsets was relevant to train the algorithm to handle files with distinct durations. The breathing phases of the third subset were manually annotated by two experts: Annotator 1 and a general practitioner and experienced lung sound researcher (H. Melbye, Annotator 3). Using Praat software (P. Boersma), annotators were able to listen to the sound while visualizing a grey-scale spectrogram and able to identify the onset and end of each breathing phase. Annotator 1 identified 479 inspiration phases and 436 expiration phases, while Annotator 3 identified 499 inspiration phases and 459 expiration phases.

### 2.3. Developed Algorithm

#### 2.3.1. Data Pre-Processing

The audio is converted to a spectrogram image representation enabling the use of image-based deep learning systems for audio classification (Figure 1). For each spectrogram, we first calculated the Short Fast Fourier Transform (SFFT) using 4096 samples per segment, with an overlap of 3200 samples between each successive segments. Second, we cropped the height of the spectrogram so that we only include data points under 2000 Hz. This result in spectrograms of around 800 × 188 pixels, depending on the length of the audio. Although spectrograms only encode information in a single channel, we used a three-channel representation since the object detection system is pre-trained using three channels.

#### 2.3.2. Object Detection

To automatically detect breathing phases we adapted the well-known Faster R-CNN (FasterRCNN) object detection system [15]. This system utilizes two convolutional neural networks, a Region Proposal Network (RPN) and a classification network. The RPN is responsible for identifying potential objects and its bounding boxes. Other relevant object proposals methods in the literature are selective search [16], sliding window approaches [17] generating windows based on edge detection, color and superpixels. We used 2000 object proposals as the default, but this can typically be tuned to suit the problem. The classification network takes the input image and classifies each of the 2000 proposals. The classification network is responsible for finding the “best” proposals. Object detection is different than image classification in that there are multiple objects and the location of the objects are important. The inspiration and expiration phases are treated as objects of different classes and there are typically multiple objects of multiple classes in each sample.

We made three changes to Faster R-CNN from reference implementation. First, we used convolutional layers from the ResNet101 architecture [18], which have been pre-trained on ImageNet [19]. Second, we changed the number of output neurons in the classification layer from 21 to 3. In the reference implementation there are 21 different classes from the PASCAL VOC dataset [20]. We used 3, representing three possible classes: background, inspiration and expiration. We tuned two hyperparameters used during the training of network. Learning rate was reduced from 0.001 to 0.0001. A learning rate >0.0005 did not converge. The number of iterations for training was increased from 70.000 to 300.000.

#### 2.3.3. Post-Processing

The object detection method outputs several proposed breathing phases and a confidence value for each one. We prune away proposed breathing phases that were below 50% in confidence. This results in 5262 removed phases from the test set (subset 3). In post-processing we used two assumptions. First, there should not be multiple detections for the same breathing phase. Multiple detections occur when the algorithm finds several phases that overlap with >50%. Post-processing removes phases that overlap and keeps the one with the highest confidence. For the test set, there were only nine such removed overlapping phases. The second assumption was that small overlaps between two successive phases were just small errors. Typically, the overlap was less than 10%. The algorithm will find these phases and correct them, in the test set there were 104 such phases. To remove the overlap, we shrink the phases in equal amounts until they no longer overlap. This step by step process is illustrated in Figure 2.

### 2.4. Evaluation of the Algorithm

In the absence of a ground truth (e.g., breathing phases detected from airflow signal), we compared the breathing phases identified by the algorithm with the ones identified by the two expert annotators (subset three). A two-step evaluation was conducted.

#### 2.4.1. Evaluation Method 1

We used boxes (Figure 2) to calculate the percentage agreement between each annotator and the developed algorithm and between annotators. We calculated the Jaccard index for all pairs of boxes (Annotator 1 vs. algorithm; Annotator 3 vs. algorithm; Annotator 1 vs. Annotator 3) and defined agreement when the Jaccard index was larger than 0.5 and the boxes were of the same class. This method emphasized the general agreement in correctly identifying the breathing phase present, and was not concerned with the agreement in detecting breathing phase bounds (i.e., the exact beginning and end of each inspiration/expiration).

#### 2.4.2. Evaluation Method 2

As usual measures of inter-rater agreement do not apply in our data where “ratings” are a function of time, we developed some continuous-time analogies of familiar measures. This allowed us to assess the performance of our method independently of discrete thresholds, but making the precise bounds of annotations count in the score. We considered the breathing phase annotation (inspiration or expiration) of a human annotator to be the regions (i.e., time periods) defined by the set A and the automatic predictions to be the regions defined by the set B. We defined true positives as the set TP=A∩B, false positives as the set FP=A−B, true negatives as the set TN=¬A∩¬B, and finally false negatives as the set FN=¬A−¬B. Here the operator ¬ is the set complement, as Figure 3 illustrates. Using the measures of these sets, we defined sensitivity (TP/(TP + FN)), specificity (TN/(TN + FP)) of the algorithm for each complete sound file against each annotator. Combining the results for inspiration and expiration and both comparisons (Annotator 1 and 3), an average sensitivity and specificity is also presented.

We used a continuous-time analogy of Cohen’s kappa, that we will refer to as pseudo-kappa. Kappa is usually defined as $\frac{p_{o}-p_{e}}{1-p_{e}},$ where $p_{o}$ is the observed agreement between raters, and $p_{e}$ is the probability of chance agreement. Agreement was defined in terms of the sets defined above; and we calculated the probability of chance agreement by calculating the agreement between files chosen at random, hence breaking the correlation between annotation and sound structure. We did this several times and took the average agreement as the probability of chance agreement. We interpreted pseudo-kappa as follows: 0 no agreement, 0–0.20 slight agreement, 0.21–0.40 fair agreement, 0.41–0.60 moderate agreement, 0.61–0.80 substantial agreement, and 0.81–1.0 almost perfect agreement [21]. For confidence intervals we relied on non-parametric bootstrap percentile intervals to account for our slightly novel use of the kappa [22].

## 3. Results

### 3.1. Evaluation Method 1

The results of Table 2 show the percentage agreements among both annotators and the developed algorithm using evaluation method 1. The method achieved a mean agreement of 97% for inspiration (between 95% and 98%) and 87% (between 79% and 96%) for expiration. There is generally a higher agreement for both inspiration and expiration between Annotator 1 and the developed algorithm, compared to Annotator 3 versus algorithm (Table 2).

### 3.2. Evaluation Method 2

When considering the fraction of time that annotators agree, all pseudo-kappa values were above 0.6 (substantial agreement). Figure 4 shows that the agreement was higher for inspiration (pseudo-kappa from 0.73 [substantial] to 0.88 [almost perfect agreement) than expiration (pseudo-kappa 0.63 [substantial] from to 0.84 [almost perfect]). Values of sensitivity and specificity of the algorithm in identifying inspiration, expiration and both breathings phases are presented in Table 3. Considering both breathing phases and the comparison with both annotators, an average sensitivity of 97% and an average specificity of 84% was found.

## 4. Discussion

To our knowledge, this is the first deep learning algorithm developed to identify breathing phases based on spectrogram image representation and using such a large lung sounds dataset (>1200 files). It is also the first attempt to utilize convolutional neural network to detect breathing phases with minimal feature engineering. The algorithm achieved an average sensitivity of 97% and an average specificity of 84%, demonstrating to be valid for detecting breathing phases in lung sounds recordings.

Our algorithm was in agreement with lung sound experts classification, 97% for inspiration and 87% for expiration, showing to be more robust in detecting inspirations in comparison with expirations. This was somewhat expected as expiration in healthy subjects can be nearly silent, being more difficult to detect from the spectrogram image. An overall good performance in identifying the breathing phases (sensitivity 97%; specificity 84%), which is in line with previous developed algorithms [4,5]. The algorithm average specificity of 84% demonstrate that the algorithm has difficulty in dealing with the parts where there is no breathing phase.

Few studies have investigated the agreement among annotators compared to an automatic method. The pseudo-kappa value above 0.60 for both the inspiration and expiration show that the agreement between each human annotator and the algorithm is comparable to the agreement between the two human annotators. Moreover, the level of agreement between the algorithm and the human annotators and between the two human annotators is similar to the level of agreement found between experts when classifying adventitious lung sounds [23]. This further shows that it can be difficult to agree where the onset and the end of the breathing phase occur in a sound recording. The annotators marked the phases by listening to the sound and marking the end when no breathing could be heard. They also read the spectrograms, which might have influenced the annotations, i.e., annotating a phase even when hardly audible breathing as it could be seen on the spectrogram. Also, the annotators did not listen to the files using the same equipment neither instructions were given regarding the volume setting for audio playback, which might also influenced the annotations. This result highlights the need for specific recommendations prior to the annotation of breathing phases in future validation studies.

We have not used standardized airflow or volume when recording lung sounds. The absence of airflow or volume signals limited the existence of a ground truth to identify breathing phases. Nevertheless, we used spontaneous airflow to increase the external validity with respect to daily clinical practice. Instead, we compared the automatic method with manual annotations from two lung sound experts. A slight better agreement of the algorithm with annotator 1 was found, in comparison with the agreement with annotator 3. This may be related to the fact that the algorithm was trained using annotations from annotator 1, and it is possible that this procedure slightly biased the algorithm towards the annotator 1. However, it should be noted that we were aiming to train our algorithm in a large dataset (~1200 files) and annotation by two experts would be very time-consuming. Moreover, annotator 1 has a vast experience in both annotation of breathing phases and adventitious sounds from recorded lung sounds, which provides confidence in the breathing phase identified. In future, it would be preferable to compare the algorithm performance with a ground truth (e.g., breathing phases detected from airflow signal) or with a multi-annotator gold-standard [13,24].

A common problem using machine learning is that the methods often work well with a contrived dataset with samples recorded in the exact same manner. In production practice, new samples will typically deviate from the constraints of the study. Using two different and large subsets to develop the algorithm we believe made the algorithm more generalizable towards new unseen datasets. In addition, the three subsets used were of different length. Subset 1 only contained files that are 10 s long, while the subset 2 and 3 contained files of 15 s. This makes the algorithm able to detect phase regardless of length up 15 s. However, going beyond 15 s would require a segmentation prior to detection. At 15 s the algorithm reaches the 11GB memory limits of the GTX1080Ti GPU during training. Also, the three used subsets come from the same dataset—the Tromsø 7 lung sound dataset, which recorded the lung sounds under the same conditions (e.g., staff, equipment, six chest locations, body position, etc.) and population (middle-aged and older adults). Our algorithm should further be validated with lung sounds acquired under distinct conditions and with other populations (including children and young adults). Our produced algorithm is available through an open access graphical user interface at https://lungsounds.medsens.io/breathing_phases. The research community is therefore invited to explore this algorithm in their own dataset and to further improve and validate the present solution.

## 5. Conclusions

We show that convolutional neural network with spectrograms as the features can be used for breathing phase detection. Our algorithm achieved an average sensitivity of 97% and an average specificity of 84%, demonstrating to be valid for detecting breathing phases in lung sound recordings. By exploring the agreement between two expert annotators and the algorithm we found that the algorithm presented here is at human-level performance. The resulting method is available through a graphical user interface at https://lungsounds.medsens.io/breathing_phases.

## Figures and Tables

**Figure 1 sensors-19-01798-f001:**
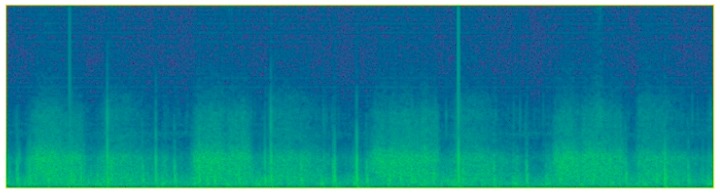
Spectrogram image representation of a lung sound recording.

**Figure 2 sensors-19-01798-f002:**
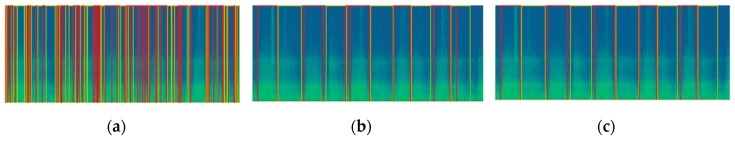
Spectrogram image representation of a lung sound recording, with red box representing inspiration and yellow box expiration phase: (**a**) without prune; (**b**) without removing overlaps; (**c**) final result.

**Figure 3 sensors-19-01798-f003:**
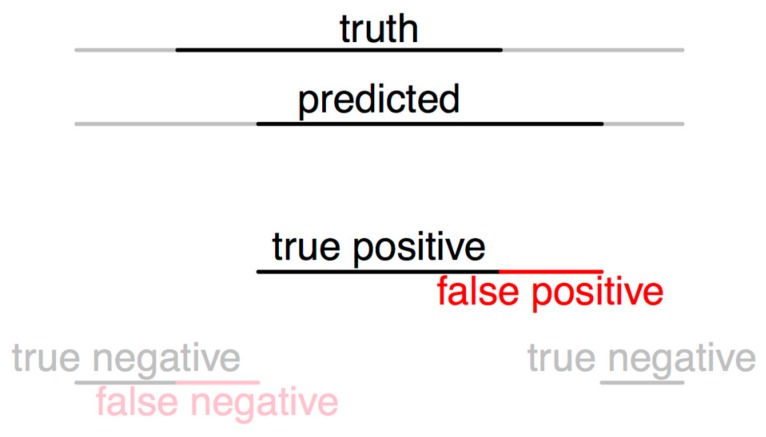
Agreement and disagreement counted as the fraction of time two annotations overlap. The black region illustrates a “positive” annotation, i.e., indicates the presence of a breathing phase; the grey region illustrates a “negative,” i.e., the absence of a breathing phase; the red region indicates the errors: annotate a breathing phase when in reality it is not present or not annotated breathing phase, when in reality it is present.

**Figure 4 sensors-19-01798-f004:**
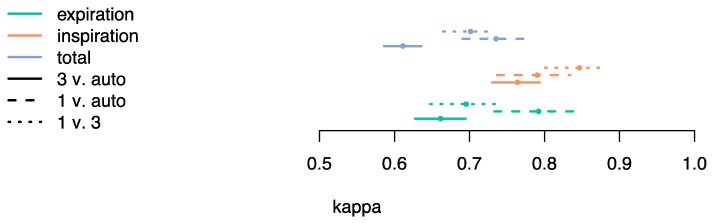
Pseudo-kappa between each annotator and the algorithm, and between annotators. Confidence intervals are of the bootstrap percentile kind.

**Table 1 sensors-19-01798-t001:** Subsets from the Tromsø 7 lung sound dataset used in this experiment.

Datasets	Annotation	N of Files	Duration	N of Inspiration Identified	N of Expiration Identified
Subset 1 (training)	Annotator 1	1022	10 s	3212	2842
Subset 2 (training)	Algorithm (inspected by Annotator 2)	112	15 s	447	418
Subset 3 (test)	Annotator 1	120	15 s	479	436
	Annotator 3	120	15 s	499	459

**Table 2 sensors-19-01798-t002:** Percentage agreements between each annotator and the automatic method using boxes.

Agreement Using Boxes	Inspiration	Expiration	Both Phases
Annotator 1 vs. Algorithm	98%	95%	96%
Annotator 3 vs. Algorithm	95%	79%	87%
Annotator 1 vs. Annotator 3	95%	84%	90%

**Table 3 sensors-19-01798-t003:** Sensitivity and specificity for both breathing phases and both annotators.

	Sensitivity			Specificity		
	Inspiration	Expiration	Both Phases	Inspiration	Expiration	Both Phases
Algorithm (Annotator 1)	97%	94%	96%	86%	87%	87%
Algorithm (Annotator 3)	98%	97%	98%	84%	78%	81%

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
