# Peer review of "Convolutional Neural Network for Breathing Phase Detection in Lung Sounds"

_sensors, 2019, doi:10.3390/s19081798_

Reviewer 1 Report

The paper explain a deep learning algorithm for breathing phase detection in lung sound recordings, with a high sensitivity  and an specificity.

It would be possible to include in Table 1, the number of inspirations and expirations not identified

On line 271, where it says GTX0180Ti, should not I say GTX1080Ti?

Author Response

Reviewer 1

The paper explain a deep learning algorithm for breathing phase detection in lung sound recordings, with a high sensitivity and an specificity. It would be possible to include in Table 1, the number of inspirations and expirations not identified

Thank you for your comment. As we do not have a ground truth (e.g., breathing phases detected from airflow signal), we do not know the total number of inspirations and expirations that were not identified. But we have now improved the discussion of this limitation. Please see page 8, lines 276-277.

On line 271, where it says GTX0180Ti, should not I say GTX1080Ti?

Thank you. This has now been corrected. Please see page 8, line 286.

Reviewer 2 Report

The authors have presented a paper on automatic detection of breathing phases in lung sounds which can have application in many areas of respiratory medicine. Overall, I believe the quality of the work presented is good. The paper is well presented, the method is sound and the results are clearly discussed. I have a few minor points that I believe should be addressed and corrected.

1) I do not agree that less attention has been given to development of techniques for automatic detection of breathing phases. A quick search on Google Scholar shows some recent methods that haven't been covered in the paper. (page 2, line 45)

2) A discussion on how methods using tracheal lung sounds differ in terms of characteristics of sounds would add more value to the paper.

3) What was the rationale behind using the number of recordings in each dataset, rather than distributing them equally?

4) Unlike subset 3, why was only one annotator used for subset 1? And a different one for subset 2? This may impact the training results.

5) One page 2, line 89: I think it should be 44,100 rather than 44.100 (i.e. not a dot after 44).

Author Response

Reviewer 2

1) I do not agree that less attention has been given to development of techniques for automatic detection of breathing phases. A quick search on Google Scholar shows some recent methods that haven't been covered in the paper. (page 2, line 45)

Thank you for your valuable comment. Indeed, the detection of breathing phases has been a topic of interest, mainly related with their possible applications to sleep apnea diagnosis and monitoring of patients. We intended to argument that methods based on lung sounds recordings until recently have received less attention. This has now been clarified (please see page 2, lines 45-48). Also we have included one more study from Reyes et al to address your concern (please see page 2, lines 59-62). Nevertheless, the strongest argument showing the innovative nature of our work continues to be that current detection methods were based on recordings from small datasets of healthy subjects and using tracheal sounds.

2) A discussion on how methods using tracheal lung sounds differ in terms of characteristics of sounds would add more value to the paper.

Thank you for your valuable comment. We have now included a discussion of the distinct characteristics of sounds acquired at trachea and chest. Please see page 2, lines 62-69.

3) What was the rationale behind using the number of recordings in each dataset, rather than distributing them equally?

Thank you for your comment. Based on previous research, samples of 100 recordings in each step would be appropriate. But in these previous experiments, recordings were mainly acquired from only one chest location. In this work we were dealing with recordings from distinct chest locations (6 in total), so higher variability across recordings was expected. Furthermore, as we were aiming to develop an algorithm based in spectrograms and we did not had previous research available on this approach, we considered that a large training dataset would make our algorithm more robust. So we targeted 200 recordings per chest location and we almost achieved that (1134 files, subset 1 and 2). For evaluation, we aimed to have at least 10% of the size of the training subset (Subset 3). Even not equitable, we believe that the three subsets used have reasonable size. This topic is now clarified in the Methods section (please see page 3, lines 104-107).

4) Unlike subset 3, why was only one annotator used for subset 1? And a different one for subset 2? This may impact the training results.

Thank you for your comment. The manual annotation of breathing phases is a very time-consuming task, so the authors agreed that the largest dataset (1022 files) would be annotated by only one annotator. But, it should be noted that this annotator is a physiotherapist with vast experience in both annotation of breathing phases and adventitious sounds in recorded lung sounds. For the validation part, as we were dealing with only 120 files, we implemented a more rigorous procedure, and two annotators were involved. But you are indeed right that this methodological choice possibly impacted the performance of the developed algorithm. To better address your concern, we have now improved the discussion of this topic in the discussion section. Please see page 7, lines 272-276.

5) One page 2, line 89: I think it should be 44,100 rather than 44.100 (i.e. not a dot after 44).

Thank you for your comment. This has been modified. Please see page 3, line 97.
